# The Impact of the Desugarization Process on the Physiochemical Properties of Duck Egg Mélange Powders

**DOI:** 10.3390/foods14091469

**Published:** 2025-04-23

**Authors:** Svetlana Kamanova, Bakhyt Shaimenova, Linara Murat, Saule Saduakhasova, Dina Khamitova, Marat Muratkhan, Baltash Tarabayev, Gulnazym Ospankulova

**Affiliations:** Department of Food Technology and Processing Products, Technical Faculty, Saken Seifullin Kazakh Agrotechnical Research University, Zhenis Avenue, 62, Astana 010000, Kazakhstan; kamanovasveta@mail.ru (S.K.); bshaymenova@mail.ru (B.S.); linaraazamatkyzy@mail.ru (L.M.); saule_aru@list.ru (S.S.); dina.khamitova@nu.edu.kz (D.K.); marat@nwafu.edu.cn (M.M.); tarabaev50@mail.ru (B.T.)

**Keywords:** duck egg, mélange, powder, desugarization, glucose, vitamins, minerals, fatty acids, amino acids

## Abstract

Duck eggs are rich in essential nutrients, such as amino acids, vitamins, and polyunsaturated fatty acids. However, their application in the food industry is hindered by glucose, which contributes to undesirable darkening during the Maillard reaction in processing. The present study examined the effect of the desugarization of duck eggs using baker’s yeast on their chemical composition. The results showed that the desugarization process reduces the content of glucose and minerals (Cu, Fe, and Zn) and alters the vitamin composition depending on the treatment conditions. Changes were also observed in the fatty acid profile, including increased levels of oleic acid (C18:1), palmitoleic acid (C16:1), and linoleic acid (C18:2, ω − 6). A high intragroup correlation among saturated fatty acids indicates the stability of their distribution. An increase in the content of essential amino acids—glycine, leucine, valine, and phenylalanine—was also recorded. Correlation analysis of the amino acid composition revealed significant relationships among both essential and non-essential amino acids. Overall, the desugarization process using baker’s yeast not only improves the nutritional profile of duck egg powder but also enhances its functional properties, positioning it as a promising ingredient for the food processing industry.

## 1. Introduction

Eggs from domesticated birds play a significant role in the human diet and are commonly consumed worldwide. In particular, duck eggs rank second in global consumption among poultry eggs. According to the China National Research System for Waterfowl Industry Technology, the annual global production of duck eggs is approximately 4 million tons, with over 90% of consumption in Asia [1].

Fresh eggs generally have mild flavors but are nutrient-dense, containing amino acids, sugars, and lipids that lead to the formation of volatile compounds [2]. Duck eggs provide a rich source of essential nutrients, such as vitamins, amino acids, and polyunsaturated fatty acids. These nutrient components fulfill basic dietary needs and contribute to the prevention of several chronic diseases [3].

Duck eggs exhibit a high lipid content, accounting for 11.40% to 13.52% of their total weight [4]. These lipids consist of 62% triglycerides, 33% phospholipids, and less than 5% cholesterol. The unsaturated fatty acids found in egg yolks are noted for their antioxidant properties, potential anticancer effects, and their role in mitigating the risk of cardiovascular diseases [5,6].

Eggs are rich in A, D, E, K, and B group vitamins and in unsaturated fatty acids, cholesterol, choline, iron, calcium, phosphorus, selenium, and zinc [7,8]. Additionally, peptides from egg whites that possess antioxidant properties have also been identified [9].

Dried eggs are widely used for their ease of storage and preparation [10], ensuring adequate stability and an extended shelf life. The main challenge in producing dried egg powders for the food industry is the browning of glucose (0.4 g/100 g) in raw eggs [11].

During powder production, glucose reacts with amino groups in a Maillard reaction, making a desugarization process necessary before drying [12,13]. As a result, the desugarization process gives egg powders additional functional properties, including emulsifying, foaming, gelling, and increased storage stability [14,15].

The main methods currently used in the egg industry for removing glucose (sugar) are natural fermentation, bacterial fermentation, yeast fermentation, and treatment with glucose oxidase [16,17,18]. Desugarization of whole egg (mélange) using baker’s yeast at a concentration of 0.2–0.4% by weight at a temperature of 22–23 °C allows for the reduction of glucose content within 2–4 h [19]. Due to the high cost of enzyme preparations, desugarization using baker’s yeast *Saccharomyces cerevisiae* is the most affordable option for egg powder producers.

Numerous studies have been conducted on the desugarization of egg whites using baker’s yeast [16,20], mostly focused on the organoleptic and rheological properties of the resulting powders. However, desugarization can also affect the quality of egg powders, which can be assessed by studying the physicochemical properties of the final products. The aim of this study is to determine the effect of the desugarization process using baker’s yeast (*Saccharomyces cerevisiae*) on the mineral, amino acid, fatty acid composition, and the content of vitamins and carbohydrates in egg mélange (whole egg).

## 2. Materials and Methods

### 2.1. Samples and Material

The study utilized fresh “Pekinskaya” duck eggs purchased from a farm in the Karaganda region, Kazakhstan. The eggs were refrigerated at 4 °C for three days before the experiment. Reagents CH_3_I, NaBH_4_, DMSO, and standard monosaccharides (galacturonic acid (GalA), glucose (Glc), rhamnose (Rha), glucuronic acid (GlcA), xylose (Xyl), fructose (Fuc), mannose (Man), arabinose (Ara), and galactose (Gal)), which were used for determining carbohydrate composition, along with vitamin and fatty acid standards, were sourced from Sigma-Aldrich (St. Louis, MO, USA). Trifluoroacetic acid (TFA) and n-hexane (chromatographic grade) were purchased from Merck (Zdarmsta, DT, Germany). The study was conducted using potassium dihydrogen phosphate 99.5% (Tokyo Chemical Industry Co., Tokyo, Japan); standard products for copper, magnesium, iron, zinc, manganese, calcium, and phosphorus (1000 μg/mL, National Nonferrous Metals and Electronic Materials Analysis and Testing Center, Beijing, China); and standard sodium chloride (purity > 99.95%∼100.05%, Tianjin Fuchen Chemical Reagent Factory, Tianjin, China). All reagents used in this study were of analytical grade.

### 2.2. Desugarization of Duck Eggs

The preparation of egg mélange and its desugarization using dry baker’s yeast *Saccharomyces cerevisiae* with experimental modification was carried out according to a previous method [21]. The control treatment was the mélange untreated with yeast. The eggs were manually cracked, and the chalazae were removed. To achieve a homogeneous mixture, the egg whites and yolks were homogenized using a High Speed Dispersator DG360 homogenizer (Changzhou, China) at 4000 rpm for 3 min. Then, a specific concentration (3.6 g/L, 3.8 g/L, or 4.0 g/L) of baker’s yeast *Saccharomyces cerevisiae* (Pakmaya, Ankara, Turkey) was added to 100 mL of egg mélange. The desugarization process was conducted at pH 7.6 and a temperature of 30 °C in an incubator for 5, 6, and 7 h. The process was halted by cooling, and the yeast cells were separated by centrifugation with a Centrifuge 5810R model Eppendorf AG 22331 (Hamburg, Germany) at 4000 rpm and +4 °C for 10 min. During this process, the yeast cells settled at the bottom of the tube, and the supernatant liquid (treatment sample-mélange) was separated by pipetting. The obtained samples of treatment were labeled as follows: 3.6/5, 3.6/6, 3.6/7, 3.8/5, 3.8/6, 3.8/7, 4.0/5, 4.0/6, and 4.0/7, where 3.6, 3.8, and 4.0 represent the concentration of baker’s yeast (g/L), and 5, 6, and 7 denote the desugarization time (hours).

### 2.3. Freeze-Drying

Freeze-drying was performed using the SB-2 sublimator (SX Technika, Kazan, Russia). To reduce the drying time, the duck egg mélange was initially placed in a freezer and blast-frozen at −40 °C until it reached −30 °C. The freeze-drying process was conducted at a desublimation temperature of −40 °C, while the shelf temperature was maintained at 35 °C during the final drying phase. The freeze-drying process lasted 14 h, with the chamber pressure ranging from 50 to 100 kPa.

### 2.4. Physiochemical Properties

#### 2.4.1. Moisture Content

The moisture percentage was estimated according to standard methods 920.116, 938.06, and 320.117 mentioned in (AOAC, 2000) [22]. The moisture content was determined by drying the treatment samples in a hot air oven at 100 ± 5 °C until a constant weight was achieved.

#### 2.4.2. Ash Content

Ash content was determined according to the method described in [22]. A 2 g treatment sample was weighed in a porcelain crucible and incinerated at 600 °C for 6 h in a muffle furnace (SnolThern, Utena, Lithuania) until white ash was obtained.

#### 2.4.3. Fat Determination

The total fat content was analyzed following the method outlined in [22]. A 3 g treatment sample was extracted using petroleum ether in Soxhlet extraction apparatus SOXTEHRM SOX414 (Gerhardt, Germany) for 6 h.

#### 2.4.4. Protein Content

The mass fraction of total nitrogen in the treatment sample was determined using the Kjeldahl method [23]. The treated 1 g of duck egg mélange powders underwent treatment with sulfuric acid, resulting in the formation of ammonium salts. Ammonium was converted into ammonia using alkaline digestion, and the ammonia was subsequently distilled and quantified using the titrimetric method.

#### 2.4.5. Solubility

The solubility of the duck egg mélange powder was identified using the modified Gao method [24]. A solution with a concentration of 10 mg/mL was prepared and stirred with a magnetic stirrer for 30 min at room temperature. Then, it was centrifuged at 6000 rpm for 20 min. The solubility was calculated using the following formula:Solubility (%) = (A/B) × 100%(1)
where A is the nitrogen content in the supernatant, and B is the nitrogen content in the mélange solution before centrifugation

### 2.5. Carbohydrates

The carbohydrate composition in duck egg powder was determined using the method in [25] with an Agilent 1260 liquid chromatograph (Santa Clara, CA, USA). The chromatographic conditions were as follows: ZORBAX SB-C18 column (4.6 × 150 mm, 5 μm), utilizing mobile phases A (50 mmol KH_2_PO_4_) and B (acetonitrile) at a ratio of 82:18. The detection wavelength was set at 250 nm, the column temperature was maintained at 30 °C, and the flow rate was 1 mL/min, along with an injection volume of 10 μL. In brief, 50 mg of powder was hydrolyzed with trifluoroacetic acid (TFA) at 120 °C for 3 h. The solution was filtered and evaporated to dryness at 65 °C. The residue was washed with ultrapure water until the odor of TFA was eliminated. To 500 μL of the resulting solution, 500 μL of 0.3 M NaOH was added, followed by 500 μL of 0.5 M PMP solution. The mixture was incubated in a water bath at 70 °C for 90 min. After extraction, 500 μL of 0.3 M HCl and 1 mL of chloroform were added, followed by shaking and centrifugation at 12,000 rpm for 5 min. The resulting supernatant was filtered using a membrane filter with a pore size of 0.22 μm.

### 2.6. Mineral Composition

The mineral composition of duck egg mélange powder was analyzed according to [26] using a NexIon 1000G inductively coupled plasma mass spectrometer (Perkin Elmer, Waltham, MA, USA).

Briefly, 300 mg of the sample was weighed, and 7 mL of concentrated nitric acid was added to a microwave digestion tube. Following digestion, the acid volume was reduced to 1 mL, and the final volume was adjusted to 50 mL for detection. The sample mass and final volume were optimized using the ICP-OES system for automatic sampling, and results were reported in mg/kg.

### 2.7. Vitamins

Vitamins were determined according to the methods in [27,28]. For vitamins A, D, and E, a 200 mg sample was weighed, and 0.5 g of ascorbic acid and 15 mL of ethanol were added. The mixture was thoroughly shaken until the ascorbic acid was fully dissolved. Then, 5 mL of a 20% KOH solution was added. The container was flushed with nitrogen to displace air, sealed with a cap, and placed in a water bath oscillator at 55 °C for 60 min to allow the reaction to occur. After cooling, the reaction mixture was transferred to a separatory funnel. Then, 20 mL of n-hexane was added, and the mixture was shaken for 10 min and allowed to stand for phase separation. The upper organic layer was collected, and the aqueous phase was extracted twice more with 20 mL of n-hexane. The organic layers were combined, washed with water until neutral, and dried using pre-dried anhydrous sodium sulfate. The extract was concentrated almost to dryness using a rotary evaporator at 40 °C. The residue was dissolved in 2 mL of methanol using ultrasonic vibration.

For vitamins B1, B2, and B3, a 1 g sample was weighed into a centrifuge tube, and 20 mL of a 0.1% formic acid solution was added. The mixture was incubated in a water bath at a constant temperature of 45 °C for 2 min. Then, 5 mL of n-hexane was added, and the sample was extracted using ultrasound for 20 min. The mixture was centrifuged at 15 °C and 10,000 rpm for 10 min. A 5 mL aliquot of the clarified middle layer was taken and filtered through a 45 µm aqueous membrane filter. From this solution, 200 µL was taken and mixed with 800 µL of ethanol. The mixture was centrifuged using a vortex mixer, and the supernatant was collected and filtered through a 0.22 µm membrane for analysis.

Mass spectrometric analysis was performed using a triple quadrupole mass spectrometer AB SCIEX QTRAP 5500 (Foster City, CA, USA). The column for liquid chromatography was a Waters C18 (2.1 × 100 mm, 1.8 µm); mobile phase: A—0.1% formic acid solution, B—methanol; flow rate: 0.2 mL/min; column temperature: 25 °C; injection volume: 5 µL.

Gradient elution program was as follows: from 0 to 1.00 min, A was 99%; from 1.01 to 4.00 min, A decreased from 99% to 45%; from 4.01 to 6.00 min, A was 45%; from 6.01 to 6.10 min, A increased from 45% to 99%; from 6.11 to 10.00 min, A was 99%.

Mass spectrometry parameters: ionization mode ESI+; gas pressure in the nebulizer: 40 psi; capillary voltage: 4.0 kV; desolvation gas flow rate (nitrogen): 10 L/min; desolvation gas temperature: 350 °C; collision gas: high purity nitrogen (99.999%).

### 2.8. Fatty Acids

The fatty acid composition was determined using the method outlined in reference [29], with experimental modifications. A 2 g treatment sample was weighed, and the fat was extracted using an automatic fat analyzer. To 10 µg of oil, 0.6 mL of NaOH–methanol (0.5 M) was added. The reaction was conducted at 100 °C for 5 min and then cooled. Next, 0.8 mL of a 14% BF3/methanol solution was added, and the reaction was carried out at 100 °C for 5 min, after which it was cooled. Afterward, 1 mL of n-hexane was added. The mixture was shaken for 30 s and allowed to stand, and then 2 mL of saturated physiological solution was added and shaken to mix. The upper organic solvent was transferred to a 1.5 mL centrifuge tube and centrifuged at 12,000 rpm for 15 min. Then, 700 µL of n-hexane was transferred to a tube containing the liquid phase for testing.

Instruments included a Shimadzu GCMS-QP2010 Ultra (Kyoto, Japan), a gas chromatography column DB-1701 from Agilent Technologies (Santa Clara, CA, USA), and high-purity helium (99.999%) as the carrier gas. The injection method was a splitless injection, with an interface temperature of 200 °C and an injection port temperature of 250 °C. The injection volume was 1.0 µL, and the ionization source used was electron impact (EI) with an ion source temperature of 230 °C. The solvent delay was 2 min, and data collection started at 2.5 min.

### 2.9. Amino Acid Content

The amino acid composition was determined according to the method in [30] using ultra-performance liquid chromatography (UPLC) (ExionLC AD, SCIEX, Darmstadt, Hessian, Germany) coupled with tandem mass spectrometry (MS/MS, QTRAP 5500, SCIEX). The chromatographic column used was Waters C18 (2.1 × 100 mm, 1.8 μm). The mobile phase was A—0.1% formic acid, B—acetonitrile. The flow rate was 0.4 mL/min, the injection volume was 10 μL, and the column temperature was set to 30 °C.

Briefly, 10 mg of the sample was accurately weighed and placed into a hydrolysis tube. Then, 5 mL of 6 mol/L hydrochloric acid was added. The tube was placed in ice water (0 °C) and frozen for 1–2 min. The air inside the tube was evacuated (to approximately 0 Pa), and the tube was flushed with nitrogen. This process of vacuum evacuation and nitrogen flushing was repeated three times. The tube was then sealed while filled with nitrogen. The sealed hydrolysis tube was placed in an oven at 110 ± 2 °C for 24 h. After hydrolysis, the tube was removed and cooled to room temperature. The contents were thoroughly mixed, and the tube was opened. The solution was transferred into a 25 mL volumetric flask. The hydrolysis tube was rinsed several times with distilled water, and the rinsing solutions were also transferred to the flask. The final volume was adjusted to 25 mL with water.

Next, 1 mL of the hydrolysate was pipetted into a vacuum drying oven and evaporated to dryness at 60 °C. After drying, 2 mL of 0.01 mol/L hydrochloric acid was carefully added. Once fully dissolved, the solution was filtered through a 0.22 µm microporous membrane for instrumental analysis.

### 2.10. Statistical Analysis

All measurements were performed in triplicate for each experiment. Statistical analysis of the obtained results was carried out using the software Origin 2021. Pearson correlation analysis was also performed to assess the linear relationship between the studied parameters. To compare the mean values and determine statistically significant differences between them, Duncan’s multiple range test (Post Hoc Duncan) was used in the IBM SPSS Statistics 27 software package. Significantly different mean values were denoted by different superscript letters (*p* < 0.05, N = 3).

## 3. Results

### 3.1. Physiochemical Properties

The effect of baker’s yeast concentration and desugarization time on the total moisture content, protein, fat, ash, and solubility in duck egg mélange powder was studied (Table 1).

As a result of the study, the moisture content of the powders was less than 5%, ranging from 2.2% to 3.8%, which is acceptable and within the normal range for storing freeze-dried mélange. The fat content in the dry powders after desugarization was slightly higher than that in the control, with the difference not exceeding 1%. According to a prior study [31], baker’s yeast contains a significant amount of fatty acids, which may explain the increase in the level of free fatty acids in the samples examined.

Yeasts release proteases that break down proteins into simpler peptides and amino acids [11], which can influence the properties of the mélange. Based on the research (Table 1), a slight decrease in protein content was observed depending on the concentration of baker’s yeast treatment. In the untreated mélange (control), the protein content was 39.9%. With the addition of yeast at a concentration of 3.6 g/L for 5 h, the protein content decreased to 39.6%. As time increased, this value further dropped to 37.5%. This trend of protein reduction was noted in all treated samples.

The ash content can vary during fermentation due to the formation and accumulation of different salts and compounds resulting from yeast metabolic activity. In the control sample, this value is 2.6%, which increases with the yeast concentration and desugarization time.

Egg powder is highly soluble in water, creating a homogeneous mixture that enhances its convenience in application. The experiment demonstrated that longer desugarization times lead to greater powder solubility. For instance, with a yeast concentration of 3.8 g and a desugaring time of 7 h, this value reached 96.3%.

### 3.2. Carbohydrate Content

According to research, eggs contain 1.0% carbohydrates, of which 0.4% is glucose, the primary carbohydrate that readily reacts with proteins [32], leading to changes in the quality of the final powders. In this study, the desugarization process using baker’s yeast was employed to address this issue. In previous studies, yeast was used to remove glucose from egg whites [33]. This research examined the carbohydrate content in duck egg mélange powders before and after desugarization, with the results shown in Table 2.

The results (Table 2) indicated an increase in carbohydrates followed by adding baker’s yeast. The experiment showed that total carbohydrate content increases based on yeast concentration and desugarization time. For instance, sample 3.6/5 had 1287.8 µg/g, sample 3.8/5 had 5921.2 µg/g, and sample 4.0/5 had 6309.8 µg/g. However, during the de-sugaring process for 7 h, the carbohydrate content decreases at all yeast concentrations, with minimal glucose content. Therefore, it was determined that a baker’s yeast concentration of 4.0 g/L is optimal for the de-sugaring process.

### 3.3. Mineral Content

Yeasts play a significant role in biosorption, absorbing various substances, including metals. In recent decades, yeasts have gained considerable attention as promising biosorbents [34]. The present study investigated the mineral composition of duck egg mélange powders and the impact of the yeast desugarization process on this composition. The findings in Table 3 reveal that the duck egg mélange powder is particularly rich in phosphorus (6565.5 mg/kg), calcium (2101.8 mg/kg), magnesium (404.5 mg/kg), and other essential minerals.

When determining the mineral composition after desugarization with yeast, a similar trend was observed in the analysis of carbohydrate concentration. Initially, the content of all minerals increased compared to the control treatment, followed by a subsequent decrease. The experiment revealed a significant drop in Cu content. For example, during desugarization with baker’s yeast at a concentration of 3.6 mg/kg for 5 h, the Cu content was 8.7 mg/kg; as the yeast concentration and exposure time increased, it decreased to 0.5 mg/kg. The Fe content also decreased, with the lowest level recorded for treatment 4.0/7, which measured 90.4 mg/kg, while it was 192.3 mg/kg for the control treatment. The Zn content also declined, reaching 49.4 mg/kg at a yeast concentration of 4.0 mg/kg after 7 h of desugarization. This reduction in metal contents in the duck egg mélange powder may be related to the biosorption process of metals by yeast, as noted by several researchers [34].

### 3.4. Vitamin Content

Most vitamins in the human body are obtained from external sources, and scientific evidence has shown that eggs are a rich source of various vitamins. For instance, refs. [35,36] reported that the retinol content in the yolk of duck eggs was 2.7 ± 1.01 µg/g, while [37] found that the vitamin A content in the yolk of organic duck eggs was 6.1 ± 0.70 µg/g. In our study, the vitamin A content in the duck egg mélange powder was 1.7 ± 0.07 µg/g. Notably, duck eggs lack vitamin C, since it can be synthesized from glucose [38].

It has been well-established that yeast plays an active role in the metabolism of B vitamins, including thiamine (B1), riboflavin (B2), pyridoxine (B6), niacin (B3), biotin, and folate. During the desugarization of glucose, yeast can synthesize increased quantities of these vitamins. Specifically, B vitamins such as B1, B2, B6, and B12 can be synthesized by yeast during the desugarization process [39].

Table 4 presents the vitamin composition of duck egg mélange powder both before and after the desugarization process.

The results of the study revealed that the vitamin A content varied between 1.1 µg/g for treatment 3.8/6 to 2.0 µg/g for treatment 3.8/7. The vitamin D3 content varied from 0.2 µg/g for treatment 3.8/5 to 0.8 µg/g for treatment 3.6/5. The vitamin E content in treatments ranged from 8.3 µg/g for 3.8/6 to 13.4 µg/g for 3.6/7, and treatments 3.6/7 and 3.6/6 had higher vitamin E concentrations. The vitamin B1 content ranged from 0.6 µg/g for 3.6/7 to 3.1 µg/g for 3.8/5, while the vitamin B2 content ranged from 6.9 µg/g for treatment 3.8/6 to 13.4 µg/g for treatment 3.8/5. The vitamin B3 content varies from 0.1 µg/g for 3.8/5 to 1.8 µg/g for 4.0/7. High levels of vitamin B3 are observed with treatments 4.0/7 and 3.6/7, where the vitamin B3 content reaches 1.8 µg/g and 1.44 µg/g, respectively.

It was found that during yeast desugarization over 5 h, the content of vitamin A decreased from 1.7 to 1.3 µg/g at all studied concentrations compared to the control sample. However, when the process duration reached 7 h, the vitamin A content increased, with the maximum amount observed at a yeast concentration of 3.8 g/L—2.0 µg/g. The content of vitamin D3 decreased at all yeast concentrations and across all time intervals.

During desugarization lasting 5 and 6 h, the content of vitamin E decreased compared to the control. However, when the process lasted 7 h, and the yeast concentration was 3.6 g/L, an increase to 13.4 µg/g was observed.

The investigation of B vitamins observed a decrease in vitamin B1 content compared to the control for all studied yeast concentrations and all time points. The content of vitamin B2 increased when the mélange was treated with a yeast concentration of 3.8 g/L, reaching 13.4 µg/g. It was determined that the content of vitamin B3 increased after 7 h of processing at all applied yeast concentrations.

Researchers [38] claim that yeast can synthesize vitamins B1 and B2 during glucose desugarization, but this is considered to have limited justification, as our empirical data provide contradictory evidence.

### 3.5. Fatty Acid Content

The lipids consist of a core of triglycerides and cholesterol esters, encased by a monolayer of phospholipids and proteins [40]. Notably, changes in the fatty acid composition were observed as a function of varying yeast concentrations and desugarization duration. Table 5 provides data on the fatty acid content in duck egg mélange powder before and after the desugarization process. Previous studies have shown that the lipids in egg yolks used for powder production are predominantly composed of oleic acid (18:1 n9), palmitic acid (16:0), and linoleic acid (18:2 n6) [41].

The obtained data on the fatty acid composition are consistent with scientific data presented in studies [40]. For instance, the palmitic acid (C16:0) content in sample 3.6/5 was 78,378.10 µg/g, 85,809.54 µg/g in sample 3.8/5, and 77,362.49 µg/g in sample 4.0/5. The control sample contained 77,272.39 µg/g. Furthermore, the desugarization process resulted in a notable increase in palmitoleic acid (C16:1), which ranged from 3700.93 µg/g to 5374.31 µg/g. Similarly, the levels of oleic acid (C18:1) and linoleic acid (C18:2, ω − 6) were elevated across all experimental samples compared to the control.

Pearson’s correlation analysis was performed to assess the correlation coefficients of the fatty acid composition. The correlation coefficients are illustrated in Figure 1.

The study of the fatty acid composition is presented as a heatmap displaying the Pearson correlation coefficients between the content of various fatty acids (µg/g) in duck egg powders. This method allows for the assessment of the degree of correlation between individual lipid profile components, which is crucial for the further development of functional food products based on duck eggs.

A high positive correlation was observed among the saturated fatty acids, particularly between pentadecanoic acid (C15:0) and heptadecanoic acid (C17:0) (r = 0.92), as well as between heptadecanoic acid (C17:0) and arachidic acid (C20:0) (r = 0.93). These correlations suggest common biochemical pathways and characteristic distribution patterns in the lipid composition of duck eggs. The saturated fatty acids heptadecanoic acid (C17:0), stearic acid (C18:0), and arachidic acid (C20:0) exhibit significant intragroup correlation, which aligns with their predominant presence in egg lipids.

The analysis revealed that stearic acid (C18:0) and erucic acid (C22:1) exhibit a strong positive correlation (r = 0.92). However, they do not share identical accumulation mechanisms in lipid structures, and their biosynthetic pathways differ. This correlation may be attributed to the similar synthesis pathways of these acids. Palmitoleic acid (POA—C16:1) shows a negative correlation with arachidic acid (C20:0) (r = −0.43), which may be due to differences in their biosynthetic pathways.

The correlation among polyunsaturated fatty acids (PUFAs) is also notable. Specifically, arachidonic acid (C20:4, ω − 6) and docosahexaenoic acid (DHA, C22:6) exhibit a strong positive correlation (r = 0.93), indicating their interconnected biochemical metabolism within phospholipids.

### 3.6. Amino Acid Composition

It is widely accepted that the Maillard reaction and lipid oxidation may be interconnected [42]. The formation of Maillard products associated with aminophospholipids has also been documented in egg-based products [43].

Yeasts contain various proteins categorized into four fractions: albumins, globulins, the ethanol-soluble protein fraction, and the alkali-soluble protein fraction. During fermentation, yeast-derived proteases release amino acids and bioactive peptides from proteins, contributing to the composition of the raw material [44].

Table 6 analyzes the amino acid composition in yeast-treated dry duck egg mélange powders. The results indicate a notable increase in essential amino acids, including glycine (control—7.88 mg/g, treatment 4.0/7—11.08 mg/g), leucine (control—29.76 mg/g, treatment 4.0/7—41.48 mg/g), valine (control—20.33 mg/g, treatment 4.0/7—29.3 mg/g), and phenylalanine (control—20.74 mg/g, treatment 4.0/7—29.3 mg/g), among others. This increase in amino acid content was observed across nearly all yeast concentrations.

Figure 2 presents the amino acid composition analysis as a heatmap, with correlation analysis highlighting the relationships in the distribution of amino acids. These insights are pivotal for developing functional food products and protein-enriched blends.

The correlations between essential amino acids reveal significant positive associations. Lysine and threonine demonstrate a strong correlation (r > 0.80), indicating their joint accumulation and vital roles in the protein profile of duck eggs. Valine and leucine exhibit a high correlation (r ≈ 0.96), as do valine and isoleucine (r ≈ 0.90), reflecting their shared synthesis and metabolic pathways within protein structures. A robust correlation (r ≈ 0.96) is also observed between phenylalanine and tyrosine, attributed to their metabolic interchangeability.

Among the non-essential amino acids, glutamic acid and arginine (r > 0.87) show a high positive correlation, which is explained by their involvement in nitrogen metabolism and protein synthesis. The strong correlation (r ≈ 0.90) between serine and glycine is due to their functional roles in proteins and their participation in purine base synthesis. Proline exhibits a moderate correlation with serine (r ≈ 0.87), supporting their interconnected metabolism in collagen formation.

## 4. Conclusions

The results of this study demonstrate that yeast-induced desugarization reduces the overall carbohydrate content, including glucose. During the desugarization process using baker’s yeast, a notable decrease in the concentrations of minerals such as Cu, Fe, and Zn was observed.

The vitamin content varied with changes in yeast concentration and the duration of the desugarization process, underscoring the significance of these factors in determining the product’s vitamin profile.

Changes in the fatty acid composition of the egg mixture during desugarification were dependent on both yeast concentration and exposure time. Specifically, an increase in palmitoleic acid (C16:1), oleic acid (C18:1), and linoleic acid (C18:2, ω − 6) was observed in the samples compared to the control. A high intragroup correlation for saturated fatty acids suggests a consistent distribution of these compounds in the duck egg powder samples.

Regarding amino acids, the levels of essential amino acids such as glycine, leucine, valine, and phenylalanine increased due to the desugarification process. Correlation analysis revealed significant relationships between both essential and non-essential amino acids. Notably, strong positive correlations were found between lysine and threonine, valine and leucine, and phenylalanine and tyrosine, indicating their joint accumulation and metabolic interdependence. Additionally, correlations between non-essential amino acids, including glutamic acid and arginine, as well as serine and glycine, emphasize their crucial roles in nitrogen metabolism and protein synthesis.

## Figures and Tables

**Figure 1 foods-14-01469-f001:**
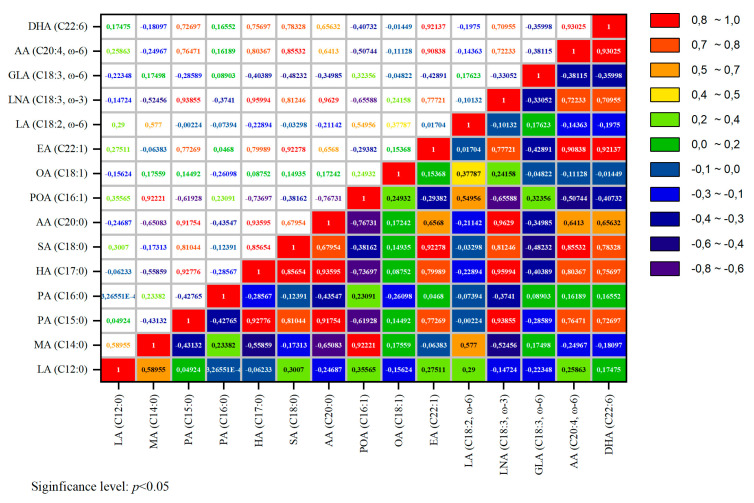
Lauric acid (LA-C12:0), myristic acid (MA-C14:0), pentadecanoic acid (PA-C15:0), palmitic acid (PA-C16:0), heptadecanoic acid (HA-C17:0), → stearic acid (SA-C18:0), → arachidic acid (AA-C20:0), palmitoleic acid (POA-C16:1), oleic acid (OA-C18:1), erucic acid (EA-C22:1), linoleic acid (LA-C18:2, ω − 6), linolenic acid (LA-C18:3, ω − 3), gamma-linolenic acid (GLA-C18:3, ω − 6), arachidonic acid (AA-C20:4, ω − 6), and docosahexaenoic acid (DHA) (C22:6). Correlation analysis of the fatty acid composition in duck egg mélange powders (positive correlation (red and orange shades r → 1), negative correlation (blue shades r → −1).

**Figure 2 foods-14-01469-f002:**
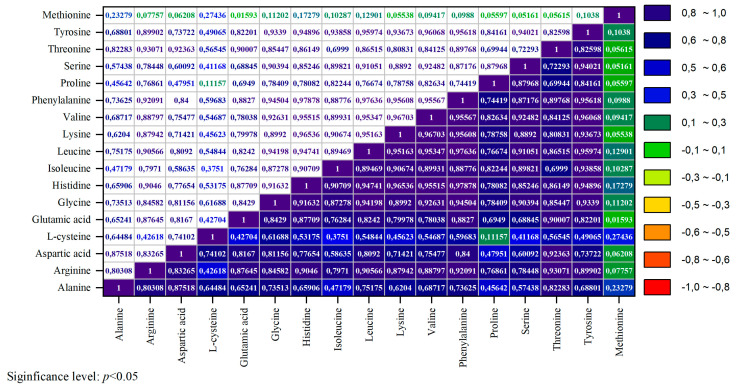
Correlation analysis of the amino acid composition in duck egg powders (positive correlation (red and orange shades, r → 1), negative correlation (blue shades, r → −1)).

**Table 1 foods-14-01469-t001:** Effects of yeast concentration and desugarization time on moisture, protein, fat, ash content, and solubility in duck egg mélange powder.

Treatment	Moisture Content, %	Fat, %	Protein, %	Ash Content, %	Solubility, %
Control	2.2 ± 0.043 ^a^	36.5 ± 0.264 ^a^	39.9 ± 0.002 ^j^	2.6 ± 0.02 ^a^	94.2 ± 1.037 ^cd^
3.6/5	3.3 ± 0.026 ^e^	36.5 ± 0.1 ^a^	39.6 ± 0.026 ^i^	2.7 ± 0.008 ^b^	92.7 ± 0.537 ^bc^
3.6/6	3.8 ± 0.029 ^f^	36.6 ± 0.305 ^a^	38.5 ± 0.09 ^g^	3.4 ± 0.007 ^c^	92.8 ± 0.771 ^bc^
3.6/7	2.8 ± 0.012 ^c^	36.7 ± 0.264 ^a^	37.5 ± 0.05 ^d^	3.4 ± 0.04 ^cd^	94.9 ± 0.141 ^de^
3.8/5	3.3 ± 0.033 ^e^	36.5 ± 0.01 ^a^	38.1 ± 0.13 ^f^	3.4 ± 0.016 ^c^	92.9 ± 0.848 ^bc^
3.8/6	3.4 ± 0.122 ^e^	36.6 ± 0.208 ^a^	35.6 ± 0.02 ^b^	3.4 ± 0.012 ^cd^	95.0 ± 0.0 ^de^
3.8/7	3.3 ± 0.032 ^e^	36.8 ± 0.136 ^ab^	34.0 ± 0.06 ^a^	3.5 ± 0.084 ^d^	96.3 ± 0.942 ^e^
4.0/5	2.9 ± 0.049 ^cd^	36.5 ± 0.236 ^a^	38.9 ± 0.12 ^h^	3.4 ± 0.012 ^cd^	88.2 ± 0.014 ^a^
4.0/6	2.9 ± 0.245 ^d^	36.8 ± 0.151 ^ab^	37.7 ± 0.032 ^e^	3.4 ± 0.004 ^cd^	89.2 ± 0.996 ^a^
4.0/7	2.6 ± 0.033 ^b^	37.1 ± 0.15 ^b^	35.8 ± 0.05 ^c^	3.5 ± 0.043 ^d^	92.4 ± 0.188 ^b^

The data are presented as means (N = 3) with standard deviations (±); for each studied parameter, values with different superscripts in the same column are significantly different (Duncan’s test, *p* < 0.05).

**Table 2 foods-14-01469-t002:** Contents of individual and total carbohydrate content in duck egg mélange powders depending on yeast concentration and desugarization time (µg/g).

Treatment	Mannose	N-Acetyl-D-glucosamine	Glucose	Galactose	N-Acetyl-D-galactosamine	Total Content
Control	559.2 ± 1.25 ^d^	46.3 ± 0.85 ^d^	135.2 ± 1.13 ^b^	408.3 ± 0.63 ^g^	227.8 ± 0.95 ^e^	1376.7 ± 0.96 ^g^
3.6/5	572.7 ± 0.34 ^e^	Trace	102.1 ± 0.91 ^a^	417.2 ± 0.01 ^h^	195.9 ± 1.03 ^d^	1287.8 ± 0.46 ^f^
3.6/6	1662.9 ± 2.58 ^f^	87.5 ± 1.07 ^e^	152.3 ± 0.57 ^c^	463.2 ± 1.42 ^k^	362.3 ± 0.79 ^f^	2728.3 ± 1.29 ^h^
3.6/7	81 ± 1.78 ^b^	38.8 ± 0.53 ^c^	Trace	237.0 ± 1.16 ^c^	138.3 ± 1.02 ^b^	495.1 ± 0.89 ^d^
3.8/5	4435.4 ± 3.42 ^i^	138.2 ± 1.39 ^i^	207.1 ± 0.89 ^f^	358.4 ± 0.32 ^f^	782.1 ± 1.26 ^i^	5921.2 ± 1.46 ^k^
3.8/6	2761.7 ± 1.21 ^g^	123.2 ± 0.77 ^h^	175.1 ± 0.66 ^d^	431.3 ± 1.04 ^j^	524.4 ± 1.44 ^g^	4015.6 ± 1.02 ^i^
3.8/7	182.0 ± 0.69 ^c^	33.1 ± 0.82 ^b^	Trace	421.1 ± 1.23 ^i^	158.9 ± 0.38 ^c^	795.2 ± 0.62 ^e^
4.0/5	4777.2 ± 4.76 ^j^	115.6 ± 0.67 ^g^	188.8 ± 1.37 ^e^	419.8 ± 1.51 ^i^	808.4 ± 1.45 ^j^	6309.8 ± 1.95 ^l^
4.0/6	3914.3 ± 4.35 ^h^	96.9 ± 1.14 ^f^	188.8 ± 0.45 ^e^	323.9 ± 0.86 ^e^	683.9 ± 0.99 ^h^	5207.7 ± 1.56 ^j^
4.0/7	78.5 ± 1.01 ^b^	27.5 ± 0.74 ^a^	Trace	304.8 ± 0.97 ^d^	139.6 ± 0.45 ^b^	424.3 ± 0.63 ^a^
Albumin	63.3 ± 0.94 ^a^	38.8 ± 0.72 ^c^	Trace	220.4 ± 0.65 ^b^	132.1 ± 1.25 ^a^	454.6 ± 0.71 ^c^
Yolk	61.6 ± 1.12 ^a^	27.5 ± 0.47 ^a^	Trace	205.1 ± 0.73 ^a^	140.2 ± 1.01 ^b^	434.4 ± 0.67 ^b^

Note: Values are expressed as means ± standard deviations. In the same column, data that do not share lowercase letters are significantly different, *p* < 0.05.

**Table 3 foods-14-01469-t003:** Mineral composition of duck egg mélange powders under different desugarization treatments, mg/kg.

Treatment	P	Zn	Fe	Mg	Cu	Ca	Total Composition
Control	6565.5 ± 7.9 ^a^	63.9 ± 0.3 ^e^	192.3 ± 3.4 ^d^	404.5 ± 2.9 ^b^	6.1 ± 0.2 ^f^	2101.8 ± 23.1 ^b^	9334.1 ± 6.3 ^b^
3.6/5	7054.2 ± 24.5 ^b^	52.1 ± 0.3 ^c^	203.5 ± 2.6 ^e^	414.3 ± 6.9 ^bc^	8.7 ± 0 ^g^	2172.5 ± 20.1 ^c^	9905.3 ± 9.1 ^e^
3.6/6	7346.2 ± 46.8 ^e^	51.2 ± 0.9 ^bc^	243.8 ± 3.3 ^h^	433.9 ± 2.4 ^de^	1.2 ± 0 ^c^	2198.5 ± 24.5 ^cd^	10,274.8 ± 12.9 ^j^
3.6/7	7296.7 ± 28.8 ^d^	50.6 ± 0.7 ^abc^	93.5 ± 1.4 ^a^	424.0 ± 4 ^cd^	1.2 ± 0 ^c^	2265.8 ± 8.8 ^g^	10,131.8 ± 7.3 ^g^
3.8/5	7216.0 ± 17 ^c^	55.1 ± 0.8 ^d^	212.0 ± 2.3 ^f^	379.8 ± 1.6 ^a^	3.4 ± 0.2 ^e^	1960.5 ± 28.5 ^a^	9826.8 ± 8.4 ^c^
3.8/6	7354.0 ± 24.3 ^e^	50.4 ± 1.5 ^abc^	152.8 ± 3.6 ^c^	417.8 ± 8.6 ^c^	0.9 ± 0.1 ^b^	2254.8 ± 11.8 ^fg^	10,230.7 ± 8.3 ^i^
3.8/7	7214.8 ± 11.3 ^c^	49.3 ± 0.3 ^a^	92.5 ± 0.6 ^a^	438.0 ± 1.6 ^e^	0.8 ± 0 ^b^	2244.6 ± 1.1 ^efg^	10,040.0 ± 2.5 ^f^
4.0/5	7253.5 ± 22.5 ^cd^	50.7 ± 1.5 ^abc^	222.5 ± 7.1 ^g^	418.3 ± 12.5 ^c^	1.2 ± 0.1 ^c^	2224.5 ± 19.5 ^def^	10,170.7 ± 10.5 ^h^
4.0/6	7076.6 ± 22.1 ^b^	50.4 ± 0.4 ^abc^	109.4 ± 4 ^b^	422.2 ± 12.2 ^cd^	1.4 ± 0 ^d^	2210.7 ± 6.4 ^de^	9870.7 ± 7.5 ^d^
4.0/7	7274.7 ± 26.5 ^d^	49.4 ± 0.7 ^ab^	90.4 ± 0.4 ^a^	433.8 ± 7.3 ^de^	0.5 ± 0 ^a^	2267.8 ± 39.3 ^g^	101.8 ± 12.3 ^a^

Note: Values are expressed as mean ± standard deviation. In the same column, data that do not share lowercase letters are significantly different, *p* < 0.05.

**Table 4 foods-14-01469-t004:** Vitamin composition of duck egg mélange powders under differentiated desugarization parameters, µg/g.

Treatment	Vitamin A	Vitamin D3	Vitamin E	Vitamin B1	Vitamin B2	Vitamin B3
Control	1.7 ± 0.07 ^e^	0.8 ± 0.08 ^d^	12.1 ± 0.01 ^d^	0.9 ± 0.2 ^d^	8.3 ± 0.13 ^bc^	1.2 ± 0.25 ^bcd^
3.6/5	1.3 ± 0.04 ^b^	0.8 ± 0.01 ^d^	12.5 ± 0.08 ^e^	0.8 ± 0.03 ^cd^	8.6 ± 0.01 ^c^	1.4 ± 0.22 ^cd^
3.6/6	1.8 ± 0.08 ^f^	0.6 ± 0 ^c^	12.6 ± 0.19 ^e^	0.7 ± 0.11 ^abcd^	7.9 ± 0.14 ^b^	1.1 ± 0.07 ^bc^
3.6/7	1.7 ± 0.01 ^e^	0.6 ± 0 ^c^	13.4 ± 0.01 ^g^	0.6 ± 0.08 ^a^	7.9 ± 0.15 ^b^	1.4 ± 0.12 ^d^
3.8/5	1.3 ± 0.01 ^c^	0.2 ± 0.01 ^a^	9.3 ± 0.03 ^b^	3.1 ± 0.15 ^e^	13.4 ± 0.1 ^f^	0.1 ± 0.02 ^a^
3.8/6	1.1 ± 0.07 ^a^	0.6 ± 0.02 ^c^	8.3 ± 0.25 ^a^	0.8 ± 0.07 ^bcd^	6.9 ± 0.16 ^a^	0.9 ± 0.06 ^b^
3.8/7	2.0 ± 0.05 ^g^	0.7 ± 0 ^d^	12.1 ± 0.29 ^d^	0.6 ± 0.01 ^ab^	8.7 ± 0.66 ^cd^	1.5 ± 0.12 ^de^
4.0/5	1.5 ± 0.01 ^d^	0.6 ± 0.01 ^c^	12.6 ± 0.18 ^e^	0.8 ± 0.06 ^cd^	9.0 ± 0.17 ^de^	1.2 ± 0.23 ^bcd^
4.0/6	1.2 ± 0.01 ^b^	0.6 ± 0.01 ^c^	10.8 ± 0.31 ^c^	0.8 ± 0.09 ^cd^	9.4 ± 0.15 ^e^	1.4 ± 0.18 ^cd^
4.0/7	1.9 ± 0.11 ^g^	0.5 ± 0.06 ^b^	12.9 ± 0.13 ^f^	0.7 ± 0.02 ^abc^	8.4 ± 0.06 ^c^	1.8 ± 0.21 ^e^

Note: Values are expressed as mean ± standard deviation. In the same column, data that do not share lowercase letters are significantly different, *p* < 0.05.

**Table 5 foods-14-01469-t005:** Fatty acid composition (µg/g) of duck egg mélange powders subjected to different desugarization treatments.

Treatment	Lauric Acid (C12:0)	Myristic Acid (C14:0)	Pentadecanoic Acid(C15:0)	Palmitic Acid (C16:0)	Heptadecanoic Acid (C17:0)	Stearic Acid (C18:0)	Palmitoleic Acid (C16:1)	Arachidic Acid (C20:0)	Oleic Acid (C18:1)	Erucic Acid (C22:1)	Linoleic Acid (C18:2, ω − 6)	Linolenic Acid (C18:3, ω − 3)	Gamma-Linolenic Acid (GLA) (C18:3, ω − 6)	Arachidonic Acid (C20:4, ω − 6)	Docosahexaenoic Acid (DHA) (C22:6, ω − 3)
Control	124.96 ± 2.7 ^h^	1059.02 ± 39.4 ^bcd^	462.62 ± 6.3 ^e^	77,272.39 ± 368.4 ^d^	1366.38 ± 124.76 ^d^	16,025.25 ± 437.66 ^g^	3672.79 ± 50.02 ^a^	426.38 ± 3.64 ^cd^	139,396.34 ± 374.67 ^a^	535.42 ± 12.1 ^e^	162.56 ± 9.09 ^a^	32,927.85 ± 17.04 ^d^	1004.72 ± 4.41 ^cd^	3556.9 ± 89.25 ^f^	556.41 ± 10.08 ^e^
3.6/5	96.58 ± 0.02 ^f^	1104.27 ± 4.14 ^def^	448.39 ± 3.55 ^de^	78,378.1 ± 434.23 ^e^	1282.48 ± 11.49 ^cd^	15,418.21 ± 41.6 ^f^	4085.34 ± 47.97 ^c^	434.44 ± 0.44 ^d^	153,311.56 ± 264.56 ^bc^	526.38 ± 2.44 ^de^	232.91 ± 1.46 ^e^	34,952.71 ± 101.93 ^f^	1016.1 ± 3.61 ^de^	3182.59 ± 1.36 ^e^	479.97 ± 0.37 ^cd^
3.6/6	91.37 ± 1.56 ^e^	1129.31 ± 25.85 ^ef^	471.79 ± 4.69 ^ef^	84,921.61 ± 349.98 ^g^	1315.75 ± 22.81 ^cd^	16,053.87 ± 108.9 ^g^	4308.34 ± 18.71 ^d^	473.13 ± 11.47 ^e^	157,777.68 ± 269.1 ^bc^	573.21 ± 0.03 ^f^	204.66 ± 1.24 ^cd^	35,714.73 ± 671.82 ^g^	1037.79 ± 12.84 ^e^	3564.81 ± 13.45 ^f^	611.95 ± 20.01 ^f^
3.6/7	112.83 ± 2.56 ^g^	1132.83 ± 37.29 ^f^	487.72 ± 20.79 ^f^	46,611.64 ± 886.21 ^a^	1303.29 ± 4.34 ^cd^	15,640.97 ± 20.47 ^f^	4349.37 ± 8.03 ^d^	505.77 ± 18.74 ^f^	172,202.25 ± 761.78 ^c^	515.63 ± 4.3 ^d^	218.68 ± 11.95 ^de^	36,213.11 ± 205.25 ^h^	992.48 ± 17.5 ^c^	2793.6 ± 70.66 ^c^	469.81 ± 13.03 ^c^
3.8/5	113.96 ± 0.79 ^g^	1279.6 ± 4.39 ^g^	284.43 ±7.18 ^a^	85,809.54 ± 429.57 ^h^	505.8 ± 13.04 ^a^	11,948.16 ± 62.73 ^a^	5374.31 ± 58.8 ^e^	25.1 ± 3.06 ^a^	142,741.71 ± 634.32 ^bc^	428.69 ± 21.54 ^a^	215.3 ± 21.54 ^d^	18,452.1 ± 7.83 ^a^	1335.08 ± 3.51 ^f^	2169.68 ± 9.27 ^a^	366.55 ± 18.52 ^a^
3.8/6	72.92 ± 0.2 ^c^	1027.98 ± 11.59 ^ab^	417.75 ± 15.77 ^c^	80,434.47 ± 235.63 ^f^	1269.59 ± 15.79 ^c^	15,089.88 ± 165.58 ^e^	3975.57 ± 48.2 ^b^	439.95 ± 1.54 d	153,330.95 ± 243.94 ^bc^	498.02 ± 9.03 ^c^	174.72 ± 2.18 ^ab^	33,260.5 ± 370.06 ^de^	957.84 ± 23.36 ^b^	2825.65 ± 62.05 ^c^	474.21 ± 13.74 ^c^
3.8/7	60.48 ± 0.22 ^a^	1045.36 ± 14.3 ^abc^	419.12 ± 20.55 ^c^	70,435.6 ± 284.79 ^c^	1090.08 ± 2.46 ^b^	12,861.6 ± 172.86 ^c^	4086.73 ± 71.7 ^c^	401.23 ± 14.24 ^b^	143,430.9 ± 243.93 ^bc^	454.35 ± 5.14 ^b^	190.45 ± 10.02 ^bc^	31,535.63 ± 33.74 ^c^	1857.34 ± 7.61 ^g^	2596.9 ± 27.83 ^b^	434.24 ± 7.32 ^b^
4.0/5	92.46 ± 0.03 ^e^	1129.84 ± 4.8 ^ef^	423.27 ± 12.42 ^c^	77,362.49 ± 264.12 ^d^	1250.46 ± 17.18 ^c^	14,726.86 ± 70.84 ^d^	4267.28 ± 12.23 ^d^	459.74 ± 1.2 ^e^	152,685.94 ± 108.28 ^bc^	537.85 ± 3.43 ^e^	174.95 ± 10.05 ^ab^	34,169.42 ± 277.59 ^e^	1033.3 ± 4.9 ^e^	3038.41 ± 16.42 ^d^	547.73 ± 13.34 ^e^
4.0/6	79.36 ± 0.34 ^d^	1083.88 ± 34.35 ^cde^	438.62 ± 4.45 ^cd^	69,945.33 ± 343.04 ^c^	1334.94 ± 27.88 ^cd^	16,158.5 ± 26.43 ^g^	4109.34 ± 42.27 ^c^	435.8 ± 3.61 ^d^	157,357.6 ± 552.24 ^bc^	533.89 ± 0.02 ^e^	174.61 ± 1.48 ^ab^	35,308.9 ± 154.6 ^fg^	967.03 ± 5.03 ^b^	3073.73 ± 41.91 ^d^	498.49 ± 4.88 ^d^
4.0/7	66.36 ± 2.45 ^b^	1005.7 ± 35.56 ^a^	394.77 ± 19.42 ^b^	68,326.51 ± 339.45 ^b^	1062.2 ± 54.79 ^b^	12,530.21 ± 108.32 ^b^	3700.93 ± 58.83 ^a^	414.2 ± 23.29 ^bc^	135,067.45 ± 382.41 ^ab^	445.88 ± 13.2 ^b^	171.86 ± 2.5 ^a^	29,486.07 ± 177.11 ^b^	821.3 ± 20.88 ^a^	2573.11 ± 30.89 ^b^	433.44 ± 9.79 ^b^

Note: Values are expressed as mean ± standard deviation. In the same column, data that do not share lowercase letters are significantly different, *p* < 0.05.

**Table 6 foods-14-01469-t006:** Amino acid content (µg/g) of duck egg mélange powders subjected to different desugarization treatments.

Treatment	Alanine	Arginine	Aspartic Acid	L-Cysteine	Glutamic Acid	Glycine	Histidine	Isoleucine	Leucine	Lysine	Methionine	Phenylalanine	Proline	Serine	Threonine	Tyrosine	Valine
Control	2.41 ± 0.06 ^ab^	15.03 ± 0.04 ^a^	7.38 ± 0 ^a^	0.57 ± 0.01 ^a^	11.7 ± 0.22 ^a^	7.88 ± 0.27 ^a^	7.61 ± 0.17 ^a^	17.71 ± 0.2 ^a^	29.76 ± 0.58 ^a^	17.4 ± 0.04 ^a^	7.47 ± 0.02 ^a^	20.74 ± 0.05 ^a^	15.47 ± 0.07 ^a^	15.2 ± 0.16 ^a^	2.36 ± 0.04 ^a^	12.17 ± 0.16 ^a^	20.33 ± 0.24 ^a^
3.6/5	2.51 ± 0.1 ^bc^	21.13 ± 0.27 ^ef^	9.04 ± 0.08 ^bc^	2 ± 0.05 ^de^	14.77 ± 0.13 ^fg^	10.03 ± 0.22 ^ef^	10.67 ± 0.1 ^e^	23.89 ± 0.08 ^g^	37.3 ± 0.91 ^d^	24.86 ± 0.01 ^e^	6.16 ± 0.12 ^e^	28.3 ± 0.72 ^ef^	18.28 ± 0.37 ^c^	19.75 ± 0.36 ^de^	3.13 ± 0 ^d^	16.74 ± 0 ^d^	26.28 ± 0.6 ^d^
3.6/6	2.55 ± 0.03 ^bc^	20.22 ± 0.06 ^de^	9.12 ± 0.09 ^bc^	1.88 ± 0.13 ^c^	13.59 ± 0.01 ^d^	9.94 ± 0.09 ^ef^	10.03 ± 0.03 ^d^	23 ± 0.1 ^efg^	38.53 ± 0.64 ^de^	25 ± 0.05 ^ef^	5.68 ± 0.05 ^d^	27.66 ± 0.11 ^e^	18.19 ± 0.19 ^c^	21.04 ± 0.07 ^f^	3.19 ± 0.08 ^d^	16.07 ± 0.14 ^c^	26.27 ± 0 ^d^
3.6/7	2.79 ± 0.2 ^cd^	21.61 ± 0.2 ^f^	9.99 ± 0.45 ^d^	2.2 ± 0.07 ^g^	14.07 ± 0.1 ^ef^	9.33 ± 0.28 ^cd^	9.84 ± 0.18 ^d^	21.58 ± 0.07 ^c^	35.88 ± 0.88 ^c^	22.57 ± 0.65 ^d^	5.44 ± 0.05 ^c^	26.73 ± 0.19 ^d^	18.25 ± 0.09 ^c^	18.92 ± 0.16 ^c^	3.55 ± 0.03 ^e^	15.69 ± 0.02 ^c^	25.41 ± 0.15 ^c^
3.8/5	2.18 ± 0 ^a^	17.05 ± 0.24 ^b^	8.38 ± 0.04 ^db^	1.97 ± 0.05 ^cd^	12.41 ± 0.13 ^b^	8.43 ± 0.01 ^b^	8.83 ± 0.01 ^bc^	19.55 ± 0.18 ^b^	32.66 ± 0.34 ^b^	21.8 ± 0.39 ^c^	4.29 ± 0.02 ^a^	24.04 ± 0.09 ^b^	15.9 ± 0.07 ^a^	16.68 ± 0.24 ^b^	2.66 ± 0.02 ^b^	13.66 ± 0.04 ^b^	23.52 ± 0.73 ^b^
3.8/6	3.16 ± 0.3 ^e^	23.05 ± 0.5 ^g^	9.66 ± 0.03 ^cd^	2.07 ± 0.08 ^ef^	13.97 ± 0.2 ^de^	10.19 ± 0.12 ^f^	10.68 ± 0.16 ^e^	23.84 ± 0.26 ^def^	38.34 ± 0.31 ^d^	26.32 ± 0.26 ^g^	6.13 ± 0.07 ^e^	28.41 ± 0.7 ^fg^	19.43 ± 0.16 ^d^	20.85 ± 0.73 ^f^	3.55 ± 0.1 ^e^	16.75 ± 0.31 ^d^	28.41 ± 0.01 ^e^
3.8/7	3.6 ± 0.04 ^f^	23.37 ± 0.21 ^h^	10.85 ± 0.1 ^e^	2.34 ± 0.05 ^h^	14.44 ± 0.2 ^fg^	9.81 ± 0.24 ^ef^	10.52 ± 0.13 ^e^	22.38 ± 0.14 ^cde^	39.7 ± 0.97 ^e^	25.43 ± 0.16 ^f^	5.8 ± 0.05 ^d^	29.09 ± 0.42 ^g^	17.66 ± 0.31 ^c^	19.31 ± 0.48 ^cd^	3.62 ± 0.08 ^e^	16.71 ± 0.42 ^d^	26.7 ± 0.49 ^d^
4.0/5	2.96 ± 0 ^de^	20.2 ± 0.93 ^dc^	8.96 ± 0.38 ^b^	2.17 ± 0.06 ^fg^	13.05 ± 0.41 ^c^	9.7 ± 0.13 ^de^	9.23 ± 0.42 ^c^	22.11 ± 0.29 ^cd^	36.39 ± 1.13 ^c^	22.71 ± 0.76 ^d^	4.63 ± 0.05 ^b^	26.04 ± 0.73 ^c^	18.15 ± 0.7 ^c^	20.6 ± 0.99 ^e^	3 ± 0.04 ^c^	16.28 ± 0.52 ^c^	25.74 ± 1.2 ^c^
4.0/6	2.76 ± 0.19 ^cd^	18.84 ± 0.31 ^c^	9.54 ± 0.94 ^cd^	1.72 ± 0.02 ^b^	13.79 ± 0.28 ^de^	9.18 ± 0.43 ^c^	8.66 ± 0.14 ^b^	19.22 ± 0.06 ^b^	33.09 ± 0.43 ^b^	20.53 ± 0.15 ^b^	4.34 ± 0.13 ^a^	24.54 ± 0.06 ^b^	16.59 ± 0.7 ^b^	16.87 ± 0.15 ^b^	3.17 ± 0.13 ^d^	13.6 ± 0.22 ^b^	22.69 ± 0.04 ^b^

Note: Values are expressed as mean ± standard deviation. In the same column, data that do not share lowercase letters are significantly different, *p* < 0.05.

## Data Availability

The original contributions presented in this study are included in the article. Further inquiries can be directed to the corresponding author.

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
