# Peer review of "The Impact of the Desugarization Process on the Physiochemical Properties of Duck Egg Mélange Powders"

_foods, 2025, doi:10.3390/foods14091469_

Round 1

Reviewer 1 Report

Comments and Suggestions for Authors

Thank you for inviting me to evaluate the Manuscript entitled "The impact of the desugarization process on the physiochemical properties of duck egg mélange powders (Manuscript ID: foods-3562091)".

This study examines the process of desugarization of duck eggs using baker's yeast and its impact on the physicochemical properties, mineral and amino acid composition, fatty acid profile, and vitamin content of egg powder.

The results offer actionable insights for mitigating the Maillard reaction in the production process of duck egg powder, which is valuable for industrial applications. However, several aspects require clarification or improvement to strengthen the scientific rigor and clarity of the work. Some suggestions are given below:

Question 1: Tables 2 and 3 lack statistical significance analysis.

Question 2: Page 6, lines 216-219. The experiment only studied fermentation times of 5-7 hours. The statement "during the desugarization process over 7 hours" lacks basis.

Question 3: Pages 6, 212-219 mainly analyze the differences between different yeast concentrations, ignoring the impact of fermentation time.

Question 4: The number of significant figures in the table data is inconsistent.

Question 5: The content of Figure 1 on page 8 shows the fatty acid composition, which is inconsistent with the statement "Table 4 and Figure 1 show the vitamin composition of duck egg melange powder before and after the desugarization process" on page 7, lines 253-254.

Question 6: Figure 2 is missing.

Question 7: Figures 1 and 3 do not show the relationship between fermentation time and yeast concentration.

Comments on the Quality of English Language

It is noted that your manuscript needs careful editing by someone with expertise in technical English editing paying particular attention to English grammar, spelling, and sentence structure so that the goals and results of the study are clear to the reader.

The authors must have their work reviewed by a proper translation/reviewing service before submission; only then can a proper review be performed. Most sentences contain grammatical and/or spelling mistakes or are not complete sentences.The quality of English needs improving.

Author Response

Dear Reviewer,

First, we would like to thank you for the trust you have placed in our work and for your dedicated time to identifying issues. We appreciate your suggestions for improving our manuscript, which we have diligently followed. Thank you for acknowledging our manuscript, which underwent the review process in the prestigious Foods journal, and for your kind guidance and timely responses.

Please kindly find our changes to the manuscript according to your comments:

Comment 1: “The results offer actionable insights for mitigating the Maillard reaction in the production process of duck egg powder, which is valuable for industrial applications. However, several aspects require clarification or improvement to strengthen the scientific rigor and clarity of the work. Some suggestions are given below:”

Response 1: The use of powders in the food industry is more practical than the use of fresh eggs, as it requires less storage space and has a longer shelf life. However, chemical processes such as the Maillard reaction occur during the drying of the powder, which can affect the quality of the final product. Nevertheless, in our article, we aimed to highlight the importance of the desugarization process by baker's yeast (Saccharomyces cerevisiae) on the nutritional properties of whole duck egg powder. We would greatly appreciate your support in publishing this manuscript in such a prestigious journal. We have carefully reviewed our manuscript and made substantial revisions.

Comment 2: “Tables 2 and 3 lack statistical significance analysis.”

Response 2: Thank you for pointing this out. According to the comment, Tables 2 (Line 245) and Table 3 (Line 263) have been changed, and statistical data analysis has been added.

Comment 3: “Page 6, lines 216-219. The experiment only studied fermentation times of 5-7 hours. The statement "during the desugarization process over 7 hours" lacks basis.”

Response 3: We agree. The research found that after 7 hours of desugarization, the glucose content was detected in a trace amount. We have made changes in lines 251–252.

Comment 4: “Pages 6, 212-219 mainly analyze the differences between different yeast concentrations, ignoring the impact of fermentation time.”

Response 4: We agree with this comment. The changes have been made in lines 248-254. The revised version is as follows:

The results (Table 2) indicated an increase in carbohydrates followed by adding baker's yeast. The experiment showed that total carbohydrate content increases based on yeast concentration and desugarization time. For instance, sample 3.6/5 had 2728.3 µg/g, sample 3.8/5 had 5921.2 µg/g, and sample 4.0/5 had 6309.8 µg/g. However, during the de-sugaring process for 7 hours, carbohydrate content decreases at all yeast concentrations, with minimal glucose content. Therefore, it was determined that a baker's yeast concentration of 4.0 g/L is optimal for the de-sugaring process.

Comment 5: “The number of significant figures in the table data is inconsistent.”

Response 5: We acknowledge that the figures in the tables were initially inconsistent. This is because the analyzed samples were processed in an accredited laboratory located in Yangling, China, at Northwest A&F University. The obtained data were incorporated into the research findings. A scanned copy of the official laboratory report is included in the supplementary materials. The significant figures in the table data have now been standardized to the same number of decimal places.

Comment 6: “The content of Figure 1 on page 8 shows the fatty acid composition, which is inconsistent with the statement "Table 4 and Figure 1 show the vitamin composition of duck egg melange powder before and after the desugarization process" on page 7, lines 253-254.”

Response 6: Thank you for pointing out this error. There was a typographical mistake in this piece of the manuscript, and the corrections have been made in lines 319–320.

Comment 7: “Figure 2 is missing.”

Response 7: We apologize for this inconvenience. Figure 2 is missing due to a typographical error, as the manuscript contains only two figures. Following the comment, the figure numbering has been corrected.

Comment 8: “Figures 1 and 3 do not show the relationship between fermentation time and yeast concentration.”

Response 8:  The selection of yeast concentrations (3.6; 3.8; 4.0 g/L) and desugarization times (5; 6; 7 hours) is based on the methodology presented in Sharma et al. [Sharma, H. K., Singh, J., Sarkar, B. C., Singh, B., & Premi, M. (2012). Statistical optimization of desugarization process parameters of liquid whole egg using response surface methodology. LWT-Food Science and Technology, 47(1), 208-212.]

Comment 9: “It is noted that your manuscript needs careful editing by someone with expertise in technical English editing paying particular attention to English grammar, spelling, and sentence structure so that the goals and results of the study are clear to the reader.

The authors must have their work reviewed by a proper translation/reviewing service before submission; only then can a proper review be performed. Most sentences contain grammatical and/or spelling mistakes or are not complete sentences. The quality of English needs improving.”

Response 9: We agree with this statement. In response to your comment, we have had our manuscript reviewed by a professional who revised the English, corrected grammatical errors, and improved the overall clarity of the text. Additionally, the authors have carefully proofread the manuscript to eliminate any typographical errors.

Once again, we sincerely appreciate your valuable insights and look forward to a productive collaboration.

Sincerely,

Svetlana Kamanova

Reviewer 2 Report

Comments and Suggestions for Authors

Although numerous studies have reported the use of yeast for protein deglycation, this manuscript presents a focused analysis of its impact on the overall quality of egg powders, including changes in minerals, amino acids, fatty acids, vitamins, and carbohydrates. The findings offer valuable reference data and potential guidance for future research and product development. However, several aspects require clarification and improvement before acceptance.

  1. Line 69: Please indicate the origin and purchase location of the duck eggs.
  2. Line 74: Please provide complete supplier information for reagents, including city, state, and country.
  3. Line 76: Please provide the source and supplier details of the yeast used in the study.
  4. Lines 85, 100, 120, and 135: Instrument brand and model information is missing. Please review the entire manuscript and revise accordingly, using the journal’s required format.
  5. Lines 120–124: Please specify the liquid chromatography elution program—whether it was isocratic or gradient.
  6. Lines 146–150: Please add detailed chromatographic and elution conditions as well as the mass spectrometry parameters.
  7. Line 147: The phrase “flow rate 0.2 mL/min” is same with Line 151; please revise for clarity.
  8. Lines 155 and 166: Please provide complete and standardized equipment information.
  9. Section 2.8: Include detailed GC-MS analytical conditions and parameters.
  10. Section 2.9: Add the chromatographic and mass spectrometric parameters used in this section.
  11. Lines 183, 193, and 204: Please unify the format of percentage expressions and decimal placement; ensure numerical precision and consistency throughout the manuscript.
  12. Line 215:Please ensure consistency in the number formatting and decimal places for values such as 2728.3 μg/g and 5921.17 μg/g, and also for values in Lines 226 and 227.
  13. Line 276: For the statement "The findings from our study align with these previous results", please cite specific references and include a comparison of data ranges between this study and the cited literatures.
  14. Figure 1: The yellow-colored data is difficult to read against the white background columns; please adjust the color scheme for better visibility.
  15. Figure 3: Similar visibility issues exist; please enhance clarity of the data presentation.
  16. The manuscript would benefit from more discussion and comparison with previous studies, highlighting the novelty and contributions of this work.
  17. Please explain the rationale for choosing yeast-based deglycation for duck egg quality analysis. Was this method compared with other deglycation strategies? If not, please justify the selection of this specific method.

Author Response

Dear Reviewer,

First of all, we would like to extend our sincere thanks for the time and energy you have spent reviewing our manuscript “The impact of the desugarization process on the physiochemical properties of duck egg mélange powders” and providing us with valuable suggestions and corrections. These comments helped us to improve the quality of our work, and we have made every effort to incorporate your recommendations into our manuscript. We truly appreciate your kind assistance and guidance.

Please kindly find our response regarding your insightful comments:

Comment 1: Line 69: Please indicate the origin and purchase location of the duck eggs.

Response 1: Thank you for pointing this out. In lines 71-72, we have indicated the origin and place of purchase of the eggs.

Comment 2: Line 74: Please provide complete supplier information for reagents, including city, state, and country.

Response 2: We acknowledge that we did not follow the proper formatting for listing the reagents used. The changes have been made in lines 77–83, including information about the suppliers of the reagents.

Comment 3: Line 76: Please provide the source and supplier details of the yeast used in the study.

Response 3: We agree with this comment and have provided complete information about the supplier of the baker's yeast. The supplier is Pakmaya Company, Ankara, Turkey, and was purchased locally. The changes can be found in lines 92–93.

Comment 4: Lines 85, 100, 120, and 135: Instrument brand and model information is missing. Please review the entire manuscript and revise accordingly, using the journal’s required format.

Response 4: We acknowledge that we did not follow the proper formatting for listing the equipment used. Corrections have been made regarding the information about the instruments in Lines 90, 96, 103, 118, 122, 139, 154–155, 163–164, 185–187, and 194.

Comment 5: Lines 120–124: Please specify the liquid chromatography elution program—whether it was isocratic or gradient.

Response 5: We appreciate your question regarding this matter.  The ratio of the mobile phase remains constant throughout the analysis, representing isocratic elution.

Comment 6: Lines 146–150: Please add detailed chromatographic and elution conditions as well as the mass spectrometry parameters.

Response 6: We agree with the suggestion to extend the details in the methodology section of our manuscript. The revisions have been made as follows in lines 167–172:

Gradient elution program was as follows: from 0 to 1.00 min, A was 99%; from 1.01 to 4.00 min, A decreased from 99% to 45%; from 4.01 to 6.00 min, A was 45%; from 6.01 to 6.10 min, A increased from 45% to 99%; from 6.11 to 10.00 min, A was 99%.

Mass spectrometry conditions: ionization mode ESI+; gas pressure in the nebulizer: 40 psi; capillary voltage: 4.0 kV; desolvation gas flow rate (nitrogen): 10 L/min; desolvation gas temperature: 350°C; collision gas: high purity nitrogen (99.999%).

Comment 7: Line 147: The phrase “flow rate 0.2 mL/min” is same with Line 151; please revise for clarity.

Response 7: We appreciate your comment regarding this typing error. The changes have been made, and they can be seen in Lines 165-166.

Comment 8: Lines 155 and 166: Please provide complete and standardized equipment information.

Response 8Thank you for pointing out this mistake. Detailed conditions and parameters have been included in section 2.8 Fatty acids (lines 185–191) as follows:

Instruments included a Shimadzu GCMS-QP2010 Ultra (Kyoto, Japan), a gas chromatography column DB-1701 from Agilent Technologies (Santa Clara, CA, USA), and high-purity helium (99.999%) as the carrier gas. The injection method was a splitless injection, with an interface temperature of 200°C and an injection port temperature of 250°C. The injection volume was 1.0 µl, and the ionization source used was electron impact (EI) with an ion source temperature of 230°C. The solvent delay was 2 minutes, and data collection started at 2.5 minutes.

Comment 9: Section 2.8: Include detailed GC-MS analytical conditions and parameters.

Comment 10: Section 2.9: Add the chromatographic and mass spectrometric parameters used in this

Response for 9/10: We fully agree with your suggestion. A more detailed description of the parameters for chromatographic and mass spectrometric analysis has been added to all sections starting from 2.5 Carbohydrates and 2.9 Amino acid content. The experiments were conducted in an accredited laboratory in China, so we have included all the data on the methods provided to us. The experimental protocol can be found in the supplementary materials.

Comment 11: Lines 183, 193, and 204: Please unify the format of percentage expressions and decimal placement; ensure numerical precision and consistency throughout the manuscript.

Comment 12: Line 215: Please ensure consistency in the number formatting and decimal places for values such as 2728.3 μg/g and 5921.17 μg/g, and also for values in Lines 226 and 227.

Response 11/12: We apologize for this mistake in our manuscript. The comments regarding the formatting of numbers and the expression format have been corrected throughout the manuscript and tables.

Comment 13: Line 276: For the statement "The findings from our study align with these previous results", please cite specific references and include a comparison of data ranges between this study and the cited literatures.

Response 13: We agree with your kind suggestion. Lines 312–313 have been revised following your comment. The corrected version is as follows:

 The obtained data on the fatty acid composition are consistent with scientific data presented in studies [40].

Comment 14: Figure 1: The yellow-colored data is difficult to read against the white background columns; please adjust the color scheme for better visibility.

Comment 15: Figure 3: Similar visibility issues exist; please enhance clarity of the data presentation

Response 14/15: We agree that the figures had low contrast, which made it difficult to see the research results. Under your comment, Figures 1 (line 320) and 2 (line 369) in the manuscript have been replaced with clearer versions.

Comment 16: The manuscript would benefit from more discussion and comparison with previous studies, highlighting the novelty and contributions of this work.

Response 16: Unfortunately, to date, there are only a limited number of publications on the study of the chemical composition of duck egg mélange powders. Most research focuses on the rheological properties of eggs or pickled (salted) eggs. As a result, it is not possible to conduct a comparative analysis or provide a more detailed discussion on the topic we have chosen.

Comment 17: Please explain the rationale for choosing yeast-based deglycation for duck egg quality analysis. Was this method compared with other deglycation strategies? If not, please justify the selection of this specific method.

Response 17: We appreciate the suggestion to provide more details about the research design explanation. In response to this comment, the following information has been added in lines 57–63:

Desugarization of whole egg (melange) using baker’s yeast at a concentration of 0.2–0.4% by weight at a temperature of 22–23°C allows for the reduction of glucose content within 2–4 hours [19]. Due to the high cost of enzyme preparations, desugarization using baker’s yeast Saccharomyces cerevisiae is the most affordable option for egg powder producers.

Numerous studies have been conducted on the desugarization of egg whites using baker’s yeast [16,20], mostly focused on the organoleptic and rheological properties of the resulting powders.

Once again, we would like to express our appreciation for your dedicated time and effort in our manuscript. We look forward to your productive collaboration and more valuable contributions and comments.

Sincerely,

Svetlana Kamanova

Reviewer 3 Report

Comments and Suggestions for Authors

The paper is focused on the desugarization process on the physiochemical properties of duck egg melange powders. The paper shows many unclear points, especially in the Introduction and M&M.
The use of duck eggs as table eggs and derivates is not all over the world, thus you should cite the law/Regulation that allow it. 
The description of the chemical composition of a duck egg should be homogeneous, showing the component content according to homogeneous units, avoiding %. You should indicate the carbohydrates content in the albumen and yolk.
The aim of the research is unclear.
In M&M, the description is, in general, unclear, as lacks details for each section of analysis. The experimental design is not described (control vs…). Which are the properties of the control (colour, ….)?
Some words are not clear (typing errors?)
Below, main suggestions (not all):
Row 60-66: did you consider the albumen melange? The aim of the research is not clear.
Row 69-70: which was the storage time for the eggs before analyses?
Row 71-74: unclear.
Row 76: egg melange: yolk and albumen? Baker’s yeast? Too vague, you have to give details.
Row 78: what is pf? why do you cite the pH value?
Row 79-83: what do you mean for set duration? how did you carry out the yeast separation? 
Row 85: details.
Row 169-173: the model used for the statistical analyses is not given, nor the number of observations for each treatment.
Table 1: what do you mean for sample? You have to change sample with treatment. Which are the components for obtaining 100%?
Captions are unclear: what do you mean for the same line?
Table 2: captions are not given.

Comments on the Quality of English Language

Some words are unclear, probably for typing mistakes.

In general, the quality of English language is fine.

Author Response

Dear Reviewer,

We would like to express our sincere gratitude for taking the time to thoroughly revise our manuscript, "The Impact of the Desugarization Process on the Physiochemical Properties of Duck Egg Mélange Powders". We appreciate your constructive comments, which have helped us enhance the scientific rigor and presentation of our manuscript.

Please kindly find our responses to your comments:

Comment 1: The paper is focused on the desugarization process on the physiochemical properties of duck egg melange powders. The paper shows many unclear points, especially in the Introduction and M&M.
The use of duck eggs as table eggs and derivates is not all over the world, thus you should cite the law/Regulation that allow it. 
The description of the chemical composition of a duck egg should be homogeneous, showing the component content according to homogeneous units, avoiding %. You should indicate the carbohydrates content in the albumen and yolk.
The aim of the research is unclear.
In M&M, the description is, in general, unclear, as lacks details for each section of analysis. The experimental design is not described (control vs…). Which are the properties of the control (colour, ….)?

Response 1: We have made every effort to incorporate all changes according to your recommendations in the manuscript.

Duck eggs, as table eggs and in products derived from them, are prohibited according to the Order of the Minister of Health of the Republic of Kazakhstan dated August 20, 2021, No.83 "On Approval of Sanitary Rules 'Sanitary and Epidemiological Requirements for the Production Facilities of Confectionery Products, Conditions for Production, Packaging, Transportation, Storage, Sale, Disposal, and Destruction of Confectionery Products' ", in Chapter 2, "Requirements for Confectionery Production Facilities, Conditions for Production, Packaging, Transportation, and Storage of Confectionery Products", which states:

According to paragraph 12.1, the use of eggs from waterfowl, chicken eggs with contaminated and/or damaged shells, eggs with defects, or eggs from farms affected by infectious diseases in birds is not allowed for the production of any cream, except for eggs from waterfowl for baking flour-based confectionery products.

According to paragraph 15.3, the use of eggs with contaminated and/or damaged shells, eggs with defects, eggs from farms affected by infectious diseases in birds, as well as duck and goose eggs, is not allowed. Source: https://adilet.zan.kz/rus/docs/V2100024077.

However, this work is being conducted under the program of the Ministry of Science and Higher Education of the Republic of Kazakhstan, grant number BR21882327 "Development of New Technologies for Organic Production and Processing of Agricultural Products," and the study is being carried out for scientific purposes. Duck eggs are permitted in the food industry and are widely used in Southeast Asian countries. Therefore, we believe the data from our research will be valuable to the scientific community.

Comment 2: Row 60-66: did you consider the albumen melange? The aim of the research is not clear.
Response 2: Thank you for pointing this out. Albumin is considered in the mixture since the mélange is a combination of egg white and yolk. In response to your second question, the following changes have been made to the manuscript in lines 66–68:

“The aim of this study is to determine the effect of the desugarization process using baker’s yeast (Saccharomyces cerevisiae) on the mineral, amino acid, fatty acid composition, and the content of vitamins and carbohydrates in egg melange (whole egg).”

Comment 3: Row 69-70: which was the storage time for the eggs before analyses?

Response 3: We agree with your comment. Clarifications regarding the storage duration of the eggs before analysis have been added in line 72 as follows:

“The study utilized fresh “Pekinskaya” duck eggs purchased from a farm in the Karaganda region, Kazakhstan. The eggs were refrigerated at 4°C for three days before the experiment.”

Comment  4: Row 71-74: unclear.

Response 4: Following your comment, clarifications and corrections have been made in lines 74–76 regarding the standard monosaccharides.  

Comment  5: Row 76: egg melange: yolk and albumen? Baker’s yeast? Too vague, you have to give details.

Response 5: We agree with your suggestion to enhance the clarity of the desugarization process in duck eggs. The following details were added to the manuscript in lines 86-98:

“The preparation of egg mélange and its desugarization using dry baker’s yeast Saccharomyces cerevisiae with experimental modification was carried out according to the method [21]. The control treatment was the mélange untreated with yeast. The eggs were manually cracked, and the chalazae were removed. To achieve a homogeneous mixture, the egg whites and yolks were homogenized using a High Speed Dispersator DG360 homogenizer (Chagzhou, Jiangsu, China) at 4000 rpm for 3 minutes. Then, a specific concentration (3.6 g/l, 3.8 g/l, 4.0 g/l) of baker’s yeast Saccharomyces cerevisiae (Pakmaya, Ankara, Turkey) was added to 100 ml of egg mélange. The desugarization process was conducted at pH 7.6 and a temperature of 30°C in an incubator for 5, 6, and 7 hours. The process was halted by cooling, and the yeast cells were separated by centrifugation on a Centrifuge 5810R model Eppendorf AG 22331 (Hamburg, Germany) at 4000 rpm and +4°C for 10 minutes, during which the yeast cells settled at the bottom of the tube, and the supernatant liquid (treatment sample-mélange) was separated by pipetting.”

Comment 6: Row 78: what is pf? why do you cite the pH value?

Response 6: We apologize for the typographical error in line 78; it has been removed. Regarding your second question, pH is mentioned as it is one of the parameters for conducting the desugarization process of the mélange (whole egg).

Comment 7: Row 79-83: what do you mean for set duration? how did you carry out the yeast separation?

Response 7: Thank you for your clarification. The following details have been provided in lines 94–98 of the manuscript:

“The process was halted by cooling, and the yeast cells were separated by centrifugation on a Centrifuge 5810R model Eppendorf AG 22331 (Hamburg, Germany) at 4000 rpm and +4°C for 10 minutes, during which the yeast cells settled at the bottom of the tube, and the supernatant liquid (treatment sample-mélange) was separated by pipetting.”

Comment  8: Row 85: details.

Response 8: We agree with your comment regarding providing more details about freeze-drying. The following clarification has been added in line 103:

“Freeze-drying was performed using the SB-2 sublimator (SX Technika, Kazan, Russia).”

Comment  9: Row 169-173: the model used for the statistical analyses is not given, nor the number of observations for each treatment.

Response 9: In response to your comment, please kindly find the changes that have been made in lines 200–206 of the manuscript as follows:

“All measurements were performed in triplicate for each experiment. Statistical analysis of the obtained results was carried out using the software Origin 2021. Pearson correlation analysis was also performed to assess the linear relationship between the studied parameters. To compare the mean values and determine statistically significant differences between them, Duncan's multiple range test (Post Hoc Duncan) was used in the IBM SPSS Statistics 27 software package. Significantly different mean values were denoted by different superscript letters (p < 0.05, n=3).”

Comment 10: Table 1: what do you mean for sample? You have to change sample with treatment. Which are the components for obtaining 100%?

Response 10: The total sum of the presented values does not reach 100%, which is standard practice in proximate analysis. This is explained by the fact that this study focuses only on the main components that characterize egg powders from a technological and nutritional perspective: moisture, fat, protein, and ash content. The remaining portion may include carbohydrates, phospholipids, cholesterol, vitamins, and other compounds that are not part of the technological parameters we investigated and, therefore, were not relevant for the purposes of this experiment, as described in Table 1. However, the carbohydrate analysis we performed focuses only on those carbohydrates that are suitable for yeast nutrition, which unfortunately does not allow us to accurately assess the total amount of carbohydrates in the egg powders studied.

Nonetheless, your valuable comment points to an important direction for further research, and we will certainly take it into account when planning the next stages of the work. We appreciate your attention to detail, as your remark has opened up new aspects of egg powder composition for future study.

Comment  11: Captions are unclear: what do you mean for the same line?

Response 11: Thank you for pointing out this mistake. Please kindly find changes made in the lines below the tables (lines 246, 264, 293, 353, 391). The word "line" has been replaced with "column".

Comment  12: Table 2: captions are not given.

Response 12: We acknowledge this error. According to your comment, captions have been added for Table 2 (Lines 245–246).

We appreciate your guidance in revising our manuscript once again and look forward to a productive collaboration with you.

Sincerely,

Svetlana Kamanova

Round 2

Reviewer 1 Report

Comments and Suggestions for Authors

1. Please pay attention to the language standard. Languages ​​other than English also appear in the article, such as "melange" in the keywords and "Растворимость" in Table 1.
2. Multiple data points in the article do not match those in the table, for example: the ash content of the control sample in row 232 is 2.5, which does not match 2.6 in Table 1. The total carbohydrate content of 3.6/5 mentioned in row 250 is 2728.3, which does not match 2 in Table 1287.8. The change in copper content in row 271 is different from the data in Table 3.
3. The analysis of rows 295-303 does not show the relationship between the change in vitamin content and the time of yeast addition and desugaring.

Author Response

Dear Reviewer,

We sincerely appreciate your time and effort in reviewing our manuscript and providing more suggestions. Thanks to your comments in Round 1, our manuscript has significantly improved.

Please kindly find our responses and the corresponding modifications that have been made:

Comment 1: Please pay attention to the language standard. Languages ​​other than English also appear in the article, such as "melange" in the keywords and "Растворимость" in Table 1.

Response 1: We apologize for the typographical error in Table 1. The word “Растворимость” has been corrected to “Solubility”. The term melange is widely accepted in scientific literature, and it refers to a mixture or blend, particularly of eggs. For example, the following references support its usage:

  1. Bakalivanov, S., et al. (2008). Characterization of freeze-dried egg mixture long stored after irradiation. Radiation Physics and Chemistry, 77(1), 58-63. https://doi.org/10.1016/j.radphyschem.2007.02.005
  2. Richards, J. F., & Morrison, B. C. (1968). Factors Influencing the Sampling of Frozen Egg Mixture for Percent Solids Content. Canadian Institute of Food Technology Journal, 1(4), 160-163. https://doi.org/10.1016/S0008-3860(68)74507-4
  3. Catalina, N. A., et al. (2016). Contributions to the knowledge of stability during storage for pasteurized egg mixture. Journal of Biotechnology, 231, S96-S97. https://doi.org/10.1016/j.jbiotec.2016.05.342

Comment 2: Multiple data points in the article do not match those in the table, for example: the ash content of the control sample in row 232 is 2.5, which does not match 2.6 in Table 1. The total carbohydrate content of 3.6/5 mentioned in row 250 is 2728.3, which does not match 2 in Table 1287.8. The change in copper content in row 271 is different from the data in Table 3.

Response 2: Thank you for noticing these inconsistencies. The data points in lines 269, 287, and 308 have been corrected to match the values in the corresponding tables.

Comment 3: The analysis of rows 295-303 does not show the relationship between the change in vitamin content and the time of yeast addition and desugaring.

Response 3: We agree with this comment. The section has been revised to clarify the relationship between changes in vitamin content and the duration of the desugarization process and yeast concentrations. These revisions are in lines 341-357:

It was found that during yeast desugarization over 5 hours, the content of vitamin A decreased from 1.7 to 1.3 µg/g at all studied concentrations compared to the control sample. However, when the process duration reached 7 hours, the vitamin A content increased, with the maximum amount observed at a yeast concentration of 3.8 g/L—2.0 µg/g. The content of vitamin D3 decreased at all yeast concentrations and across all time intervals.

During desugarization lasting 5 and 6 hours, the content of vitamin E decreased compared to the control. However, when the process lasted 7 hours, and the yeast concentration was 3.6 g/L, an increase to 13.4 µg/g was observed.

The investigation of B vitamins observed a decrease in vitamin B1 content compared to the control for all studied yeast concentrations and all time points. The content of vitamin B2 increased when the melange was treated with a yeast concentration of 3.8 g/L, reaching 13.4 µg/g. It was determined that the content of vitamin B3 increased after 7 hours of processing at all applied yeast concentrations.

Researchers [38] claim that yeast can synthesize vitamins B1 and B2 during glucose desugarization, but this is considered to have limited justification, as our empirical data provide contradictory evidence.

We appreciate your work in revising our manuscript and dedicating time to its improvement. If you have any further corrections or concerns, please do not hesitate to suggest them.

Sincerely,

Svetlana Kamanova

Reviewer 2 Report

Comments and Suggestions for Authors

The authors have made a sincere and thorough effort in addressing the reviewer’s previous concerns. The revised manuscript shows significant improvement in clarity, methodological transparency, and data interpretation. The current version could be considered for acceptance.

Author Response

Dear Reviewer,

We would like to express our sincere appreciation for your comments and the time you dedicated to evaluating our manuscript. Your constructive feedback during Round 1 has significantly improved the quality of our work. We wish you continued success in your professional path and all the best in your future research.

Thank you once again for your careful review and support.

Sincerely, Svetlana Kamanova

Reviewer 3 Report

Comments and Suggestions for Authors

The paper has been partially improved according to suggestions. However, it still shows unclear parts.

The year of the Grant has to be given.

Description of the chemical analyses are not homogeneous, as the size of samples (mass) has to be given for each analysis.

Statistics: number 3 (samples/treatment) is too low for realistic statistical treatment of the data.

Captions of the tables are unclear (Table 1, table 2, ...) and the graphics are not readable.

Table 2: it is unclear why albumin and yolk have been added to the table. Furthermore, the title of the table is on melange.

Conclusions does not agree (partially) with conclusions of the abstract.

Author Response

Dear Reviewer,

Thank you for taking the time to review our manuscript once again and provide your valuable comments. Your suggestions in Round 1 were clear and constructive, and they have greatly improved the clarity of our work.

We have carefully addressed each of your comments and made the recommended changes. Please find our responses below:

Comment 1: The year of the Grant has to be given.

Response 1: Thank you for pointing this out. The year of the funding period (2024-2026) has been added in Line 477.

Comment 2: Description of the chemical analyses are not homogeneous, as the size of samples (mass) has to be given for each analysis.

Response 2: We agree with this suggestion to improve the clarity of the Materials and Methods section. Following your comment, the sample mass used for each analysis has been added and is now indicated in the manuscript (Lines 124, 161–181, and 220–234).

Comment 3: Statistics: number 3 (samples/treatment) is too low for realistic statistical treatment of the data.

Response 3: We agree that increasing the number of replicates could enhance the reliability of the results. However, in this study, each treatment was analyzed in three replicates (n = 3), which is consistent with accepted standards in peer-reviewed publications in the field of food science.
Statistical analysis was performed using Origin 2021 software. Additionally, Pearson correlation analysis assessed linear relationships among the studied parameters. Duncan’s post hoc test (IBM SPSS Statistics 27) was applied to compare mean values and determine statistically significant differences. Significantly different means were indicated by different superscript letters (p < 0.05, n = 3).

Comment 4: Captions of the tables are unclear (Table 1, table 2, ...) and the graphics are not readable.

Response 4: Thank you for your observation. The table captions have been modified according to your comment. However, all figures in the manuscript are already accompanied by explanatory text and a clear indication of measurement units. Differences between samples are marked using statistical indices. We have carefully reviewed the formatting and ensured that it complies with the journal’s requirements.

If you or the Editorial Office deem it necessary, we would gladly provide the figures in a separate high-resolution file for further review or to include a link to supplementary images in the appendix.

Comment 5: Table 2: it is unclear why albumin and yolk have been added to the table. Furthermore, the title of the table is on melange.

Response 5: In response to your comment, appropriate changes have been made to Table 2.

Comment 6: Conclusions does not agree (partially) with conclusions of the abstract.

Response 6: We agree with this suggestion. We have revised the abstract to ensure agreement with the main conclusions.  The updated abstract can be found in Lines 12-21:

"The present study examined the effect of the desugarization of duck eggs using baker’s yeast on their chemical composition. The results showed that the desugarization process reduces the content of glucose and minerals (Cu, Fe, and Zn) and alters the vitamin composition depending on the treatment conditions. Changes were also observed in the fatty acid profile, including increased levels of oleic acid (C18:1), palmitoleic acid (C16:1), and linoleic acid (C18:2, ω-6). A high intragroup correlation among saturated fatty acids indicates the stability of their distribution. An increase in the content of essential amino acids—glycine, leucine, valine, and phenylalanine—was also recorded. Correlation analysis of the amino acid composition revealed significant relationships among both essential and non-essential amino acids."

Once again, we sincerely appreciate your feedback and support of our manuscript. Thank you for your revision and for sharing your ideas on how to improve our work. We wish you continued success in your research.

Sincerely,

Svetlana Kamanova
